# A Scoping Review of the Health Impact of the COVID-19 Pandemic on Persons Experiencing Homelessness in North America and Europe

**DOI:** 10.3390/ijerph19063219

**Published:** 2022-03-09

**Authors:** Julia Corey, James Lyons, Austin O’Carroll, Richie Stafford, Jo-Hanna Ivers

**Affiliations:** 1Department of Public Health & Primary Care, School of Medicine, Trinity College Dublin, D24H74 Dublin, Ireland; coreyj@tcd.ie (J.C.); lyonsj8@tcd.ie (J.L.); 2North Dublin City GP Training Scheme, D07H984 Dublin, Ireland; austin.ocarroll@icgp.ie; 3HSE Community Healthcare Organisation Dublin North City & County, D09C8P5 Dublin, Ireland; richie.stafford@hse.ie

**Keywords:** homelessness, health, COVID-19, pandemic

## Abstract

Persons experiencing homelessness (PEH) are at heightened risk for infection, morbidity, and mortality from COVID-19. However, health consequences of the pandemic extend far beyond those directly caused by the virus. This scoping review aimed to explore the impacts of the COVID-19 pandemic on the health and well-being of PEH in North America and Europe. A systematic search of academic and grey literature was conducted in September 2021. To be included, studies had to include primary data related to the impact of the pandemic on health or well-being of PEH and be written in English. All potentially relevant references were independently screened by two reviewers, and minor conflicts were settled with input of a third reviewer. A total of 96 articles met criteria for inclusion. Data extraction was completed for all included studies, and findings synthesised and presented thematically. Numerous health impacts of the pandemic on PEH were identified, including SARS-CoV-2 infection, morbidity, mortality, and hospitalisation, fear of infection, access to housing, hygiene, PPE, food, as well as mental health, substance use, other health-related outcomes and treatment services. Gaps in the literature relating to persons using alcohol, access to mental health support, and violence were also identified. Implications for future research are discussed.

## 1. Introduction

Since the emergence of SARS-CoV-2, the virus that causes coronavirus disease 2019 (COVID-19), more than 350 million people worldwide have experienced a confirmed infection, and more than 5.6 million have died [1]. Despite widespread reductions in risk of COVID-19 transmission and severe disease as a consequence of vaccine rollouts [2], the virus continues to pose a serious threat to human health, particularly for vulnerable populations [3,4]. Persons experiencing homelessness (PEH) appear to be at higher risk for infection, morbidity, and mortality due to COVID-19 than the general population [5,6,7,8,9]. Several factors may contribute to this increased risk, including lack of safe housing, inadequate access to healthcare, difficulties following public health guidelines, closure of regularly attended support services, and heightened risk for comorbidities including chronic diseases, mental health issues, and addiction [6,10].

While PEH have been disproportionately affected by the pandemic [5,8], the health consequences reach far beyond those of just infection and illness from COVID-19. For example, the large-scale closure of public buildings and facilities reduced access to toilets and basic hygiene and sanitation for those who were unsheltered [11,12]. Less financial support from the general public due to stay-at-home advisories and closure or reorganisation of food and social support services also contributed to difficulties accessing food assistance and hygiene products [12,13]. Restrictions on international travel and freedom of movement interrupted alcohol and drug supply chains, increasing costs [14] and reducing access, leading some persons who use drugs to shift from using their preferred substance to potentially more dangerous substances [15]. In some cases, harm reduction, treatment, and support services for PEH were closed, limited, or inaccessible due to commute barriers [16,17,18,19]. These new challenges have exacerbated existing mental health issues; studies have reported increased rates of self-harm, depression, and anxiety among PEH [20,21], as well as increased difficulties in accessing mental health services [13].

Despite the negative impact the pandemic has had on PEH, unprecedented changes in policies and services brought about to reduce the spread of COVID-19 have also improved the health and well-being of PEH in some communities. Positive changes have included the rapid rehousing and shielding of PEH [22,23], expansion of harm reduction and treatment services [22,24,25], and cross-sector collaboration of services that increased access to mental health services [26].

It is vitally important to sustain and improve healthcare services for PEH in the face of the growing public health threat posed by COVID-19 and beyond. While previous reviews have examined the health consequences of COVID-19 on PEH, they have largely focused on infection rates and control measures [27,28] or the pandemic response [29,30]. As part of a broader project to evaluate the policies and practices that were implemented during the pandemic to minimise the risk of SARS-CoV-2 infection for PEH who use drugs, this scoping review was conducted to explore the impacts of the COVID-19 pandemic thus far on the health and well-being of PEH in North America and Europe.

## 2. Materials and Methods

The methodology chosen for the current study was a systematic scoping review. Scoping reviews can be used to identify and map the available evidence on a topic, as well as to identify gaps in the knowledge base. They are particularly useful for exploring a topic that is diverse, complex, and under researched [31].

### 2.1. Search Strategy

A systematic search was conducted across five databases (EMBASE, Medline, CINAHL, PsychINFO, and Web of Science), and grey literature was searched using Google Scholar. Search terms were aimed at all papers encompassing PEH, cross-referenced with COVID-19. As strategies were tailored to each database, a detailed search strategy is presented in Appendix B. The authors consulted a subject librarian with expertise in health research and search strategies before carrying out the search to ensure quality and accuracy.

### 2.2. Study Selection

The literature search was conducted on 15 September 2021. Identified articles were first exported to an EndNote Library and then uploaded to the systematic review screening platform, Covidence (www.covidence.org, accessed on 3 February 2022), for removal of duplicates and screening. Articles were independently screened by two researchers (J.C. and J.L.), and minor disagreements were resolved through discussion between the researchers with input from a third researcher (J.I.). The inclusion criteria for the current review selected articles that (1) included primary data related to the impact of COVID-19 on health or well-being of PEH and (2) were written in English. Studies were excluded if (1) homelessness was defined as a history of homelessness that predated 1 January 2019; (2) the study was a simulation and did not include real-world data (i.e., mathematical models used for predictions); (3) the study was published before March 2020; (4) the study was conducted outside of North America or Europe; (5) the study published as an abstract only or there was no full-text available; (6) the article was an ethnography. The decision to exclude studies that defined history of homelessness as homelessness predating 1 January 2019 was agreed upon by the research team to ensure that included studies were based on up-to-date housing data for the populations discussed therein. Additionally, as the World Health Organisation declared COVID-19 a pandemic in March 2020 [32], publication in or after this month was deemed by the research team to be an appropriate requirement for included literature in order to capture the health impacts of the pandemic on PEH. It was not required for the included studies to focus solely on PEH, but it was required that they provide data for PEH that was disaggregated from the broader study population, where applicable, to enable the impact of COVID-19 on health or well-being to be assessed specifically for PEH. Finally, a validation check was conducted by two authors (R.S. and A.O.) with extensive knowledge on the field, who reviewed and suggested additional literature on the topic for inclusion (*n* = 1).

### 2.3. Data Charting and Synthesis

The first round of data extraction was conducted using a Covidence extraction template specifically designed by the research team to capture predetermined aspects of study design (cross-sectional, longitudinal, qualitative, mixed-methods, case study, etc.), study description (country and primary population of focus, article type), sample characteristics (participant gender/sex, age, sampling strategies), outcome measures, and results. The extraction template was independently piloted using five articles by two authors, and minor adjustments were made as necessary. In total, 16 studies were extracted independently by two authors to ensure quality and consensus on relevant data. Following consensus and discussion, the remaining studies (*n* = 80) were extracted independently by one author and cross-checked for accuracy by a second author. Upon completion of extraction, data were exported to Microsoft Excel and further analysed based on subthemes that emerged. Data are synthesised and presented according to these subthemes in the following section. Given the depth and breadth of articles included in this review, a meta-analysis was not possible. As such, the following section presents an overview of the characteristics and findings of included studies as a means of producing suggestions for future research and strengthening the evidence base in relevant areas.

## 3. Results

### 3.1. Screening Process

Figure 1 details the initial search returns of five different databases and a grey literature search, with 1554 articles identified for screening. Of these, 734 were automatically removed as duplicates, resulting in 820 articles for title and abstract screening. After screening titles and abstracts, 313 full-text studies were assessed for eligibility. One additional study was identified by two authors (R.S. and A.O.C.) with extensive knowledge in the field and assessed for eligibility. Following the review of full texts, 96 articles met the inclusion criteria for data extraction and analysis.

### 3.2. Overview of Included Studies

The majority of studies included in this review collected primary data to answer a specific research question (*n* = 85). Eight studies conducted analysis of secondary data. PEH were the primary population of focus of 74 studies, while other literature looked at PEH as a smaller subset of study populations, such as general patient populations (*n* = 6), persons living with HIV (*n* = 5), COVID-19 patients (*n* = 3), multiple vulnerable groups (*n* = 3), or others (*n* = 5), as outlined in Table 1. Most of the included studies were original journal articles (*n* = 48); however, given the time-sensitive nature of information during the current pandemic and the often lengthy process of peer-review, a number of different article types were included in the current review, including brief reports (*n* = 8), research letters (*n* = 6), preprints (*n* = 4), short communications (*n* = 4), case studies (*n* = 4), reports (*n* = 4), letter to the editor (*n* = 3), and others (*n* = 15), as detailed in Table 1. Studies were conducted in more than ten countries across North America and Europe. Most (*n* = 51) were conducted in the United States (U.S.), followed by the United Kingdom (U.K.) (*n* = 9), France (*n* = 9), and Canada (*n* = 6). Table 1 provides a full breakdown of study locations. Studies also differed by design, as presented in Table 1. Detailed data extraction from each included study can be found in Appendix A.

### 3.3. SARS-CoV-2 Infection and Morbidity

A number of included studies reported infection attack rates of SARS-CoV-2 among study participants: 29 studies measured positivity rates (i.e., current infection) [13,33,34,35,36,37,38,39,40,41,42,43,44,45,46,47,48,49,50,51,52,53,54,55,56,57,58,59,60], seven measured seroprevalence (i.e., prevalence of antibodies for SARS-CoV-2, indicating historic infection) [61,62,63,64,65,66,67], and four measured both positivity and seroprevalence rates [68,69,70,71]. Additionally, data regarding SARS-CoV-2 symptoms reported by study participants were provided by 32 articles [7,8,33,34,35,36,38,42,45,46,47,48,49,50,51,52,55,56,57,58,61,62,68,69,70,72,73,74,75,76]. Positivity rates among PEH ranged from 0 to 66%, while seropositivity among PEH ranged from 0 to 94%. Among studies that measured positivity rates, 13 included testing of participants in response to an outbreak [34,36,38,39,40,43,44,45,48,50,55,56,68]. In addition, surveillance testing (i.e., testing not prompted by a confirmed or suspected outbreak) was conducted for all participants regardless of symptoms in 16 studies [33,35,36,38,40,41,42,43,44,46,47,52,58,69,70,71]. Of these, six reported that routine surveillance testing occurred regularly [35,41,44,52,58]. Eight studies described close-contact- or symptom-based testing strategies [36,44,49,53,54,56,58,71], and one study did not specify reasons participants were tested [60]. Two studies reported positivity rates using retrospective [57] or prospective (person-level data from COVID-19 surveillance and reporting systems) [37] data from cohorts of the populations that were tested for SARS-CoV-2. One study by Jatt et al. [36] described how testing progressed from only symptomatic individuals and close contacts to outbreak testing and finally to routine surveillance testing from 11 March to 29 April 2020 in a large healthcare facility in Los Angeles. Of the 33 studies that measured infection rates, polymerase chain reaction (PCR) tests were the most commonly reported tests used for diagnosis (*n* = 24) [34,36,40,41,42,43,44,46,47,48,49,50,51,52,53,54,56,58,60,68,69,70,71,77]. Other measures included self-reported positive test results (*n* = 3) [13,59,70], antigen testing (*n* = 1) [35], and assays (*n* = 1) [33]. Five studies did not specify tests used for SARS-CoV-2 diagnosis [37,38,39,55,57].

Multiple unique risk factors for SARS-CoV-2 positivity were identified among PEH. Three studies reported that PEH were at increased risk for infection compared to the general population [37,62,78], though one study found that among people living with HIV, experiencing homelessness was not associated with seropositivity [64]. Among haemodialysis patients, Rincón et al. [45] found that living in a nursing home or experiencing homelessness was an independent risk factor for testing positive for SARS-CoV-2. Three studies reported risk of COVID-19 based on specific shelters of residence [43,47,68]. While no statistical significance was reported, Rogers et al. [44] noted that most positive cases (79%) were detected among shelters housing older male residents and with shared day services, showering facilities, and rotating staff. Similarly, Ghinai et al. [48] reported that increased numbers of private bathrooms were associated with lower prevalence rates. Living in a congregate or crowded setting [61] and shared sleeping arrangements [48] were also identified as risk factors. In addition, a study by Roland et al. [69] reported that persons who shared a room with someone who tested positive, or did not know, were significantly more likely to test positive. Rogers et al. [44] reported that 86% of positive cases in their study of homeless shelters in King County, Washington slept in a communal space in the past week, compared with 78% of residents with negative tests; however, no statistical significance was provided. Three articles noted that the shelters in their studies with the largest outbreaks had more transient resident populations [33,35,46]. Similarly, Ghinai et al. [48] reported an association between the proportion of residents leaving and returning each day and increased prevalence rates. A study of congregate shelters in Rhode Island found that 70% of participants with negative tests had spent more than two weeks at their shelter, compared with 43% of participants with positive tests [46]. The same study reported that three shelters that had stopped accepting new residents at least two weeks prior had zero cases at the time of testing [46].

Additional risk factors were identified by several studies. Two studies found the presence of symptoms to be associated with testing positive for SARS-CoV-2 [47,61], while four studies found no statistical significance between symptoms of persons testing positive and negative [33,38,46,69]. Two studies reported a relationship between older age and increased likelihood of testing positive among PEH [38,48], though this relationship was not statistically significant in the study by Kiran et al. [38], and after adjusting for individual level factors and clustering at shelters, positivity rates no longer differed significantly by age in the study by Ghinai et al. [48]. Another study by Ly et al. [47] reported that younger PEH (18–34 years old) had more than three times higher odds of testing positive. Karb et al. [46] reported no statistical differences in age between people testing positive and negative. One study reported that 84% of PEH testing positive for SARS-CoV-2 were male, though men accounted for 72% of participants [50], and three found no significant difference in gender between people testing positive or negative [46,48,69]. Prevalence of the virus was higher among non-Hispanic white PEH than among non-Hispanic Black PEH in a study by Ghinai et al. [48], though after adjusting for individual-level factors, the positivity rates no longer differed. Karb et al. [46] also reported no differences in race between persons testing positive or negative. PEH in Belgium with an Urgent Medical card had significantly higher proportions of SARS-CoV-2 infections than those without access to the health system (7% vs. 3%, respectively) [69], and shelter residents in Toronto who tested positive were significantly less likely to have a provincial health insurance card than those who tested negative (54% vs. 72%, respectively) [38]. In France, Rahi et al. [79] found that PEH were more likely to be infected during lockdown (17 March–11 May 2020) than before (5% vs. 1%, respectively). One study reported significantly lower seroprevalence among shelter residents consuming tobacco (3%) compared to those who did not (8%) [61], and another study similarly found that current smoking among shelter residents was associated with lower prevalence of infection, compared with never smoking [48]. PEH who tested positive in a study by Karb et al. [46] had significantly lower prevalence of comorbidities than those testing negative (20% vs. 40%, respectively). Seroprevalence was also reported to be lower among PEH with psychiatric and/or addiction comorbidities than among those without (3% vs. 6%, respectively) in a study by Loubiere et al. [61]. Prior chronic respiratory disease [69] and self-reported medical history [48] were not significantly associated with positivity status in two other studies [48,69].

### 3.4. COVID-19-Related Hospitalisation

Nine studies provided information regarding COVID-19 related hospitalisations among PEH [8,34,48,49,55,68,73,75,76], among which six provided hospitalisation rates [34,48,49,55,68,73]. A study by Imbert et al. [73] reported that 8% of shelter residents who tested positive between 5 April and 15 April 2020 required hospitalisation, and a study by Tobolowsky et al. [55] reported that 20% of residents that tested positive between 30 March and 1 April 2020 were hospitalised. Among symptomatic persons with COVID-19, significantly more PEH were hospitalised than those in the general population (29% vs. 11%, respectively) in a study by Fields et al. [49]. Another article reported that among the 13% of shelter residents that were hospitalised due to COVID-19 (testing positive between 1 April and 1 May 2020), 33% required intensive care unit (ICU) admission [48]. In Paris, 24% of residents across three homeless shelters that tested positive were hospitalised, of which 12% were transferred to an ICU [68]. Data from one shelter in the same study indicated that patients over 65 years old, those with heart conditions, those with chronic kidney disease, and those with more than two risk factors were hospitalised more often [68]. In a Toronto refugee shelter, 4% of residents that tested positive for SARS-CoV-2 were admitted to the hospital for isolation requirements rather than clinical severity; there were no reported cases of ICU admission at the time of the study (20–21 April 2020) [34]. A U.S. study by Cha et al. [76] reported that among patients experiencing homelessness hospitalised with COVID-19, 54% were hospitalised for >4 days, 17% were admitted to the ICU, and 11% had mechanical ventilation, most commonly patients > 65 years (20%) and those with no underlying health conditions (21%). Schrooyen et al. [8] reported that incidences of hospitalisation for COVID-19 were three times higher among PEH (650 per 100,000) compared to the general population (194 per 100,000). Among all adult patients with COVID-19 treated from 1 March to 18 May 2020 at the Boston Medical Centre (BMC), PEH accounted for 16% of all patients, 24% of non-ICU inpatients, 16% of ICU admissions without mechanical ventilation, and 15% of ICU admissions with mechanical ventilation [80]. Hospitalisations among COVID-19 positive PEH in the BMC system were reduced by 28% following the opening of the COVID-19 Recuperation Unit, located adjacent to the BMC, which provided space for PEH to isolate, quarantine, and receive treatment for substance use [75].

### 3.5. COVID-19-Related Mortality

Mortality from COVID-19 among PEH was evaluated in 12 studies [5,8,37,48,49,53,68,72,73,76,80,81]. Data regarding case fatality rates (CFRs) were available from eight studies [5,37,48,53,68,72,73,76], with six studies reporting at least one death [5,37,48,49,53,68,73,76]. A study from France by Husain et al. [68] reported the highest CFR (6%) among the included studies, using data from PEH in shelters who received positive PCR tests for SARS-CoV-2 between 1 March 2020 and 31 May 2020. Leifheit et al. [5] found that, when compared to the wider population, CFRs in Los Angeles were significantly increased for PEH under the age of 65, and that the opposite was true for those ages 65 and over. A study by Hsu et al. [80] reported that among adult patients with confirmed COVID-19 that were treated at BMC, 15% of those who died were PEH. Additionally, 15% of deaths among PEH in Wales between February and July 2020 were registered as COVID-19 involved, compared with 14% among the general population [81], though no statistical significance was provided. One study reported no significant association between housing problems and mortality from COVID-19 [8].

### 3.6. Fear of COVID-19

Fear or perceived threat of COVID-19 among PEH was discussed by ten studies [12,13,82,83,84,85,86,87,88,89]. Fear varied across studies; in Los Angeles, 65% of tenants of permanent supportive housing surveyed by Henwood et al. [85] in March 2020 regarded COVID-19 as a serious risk to their health, while 33% of those surveyed in Los Angeles by Kuhn et al. [87] from December 2020 to February 2021 perceived it as a high threat. A third study in Los Angeles carried out from April to June 2020 reported that 53% of young PEH (18–25 years old) were not at all worried about COVID-19, and 15% were very or extremely worried [88]. One study by Rodriguez et al. [12] reported that PEH in Tippecanoe County, Indiana had an overall low risk perception of COVID-19, while authors of a study in France found that PEH felt that the virus was indeed a threat but was not a major concern compared to the other risks they regularly faced [13]. Using data from Hamburg, Hajek et al. [89] found that increased fear of COVID-19 among PEH was associated with younger age, absence of chronic alcohol consumption, increased perceived own risk of contracting the virus one day, and a higher agreement that a diagnosis of COVID-19 would ruin their life. Similarly, Henwood et al. [85] reported that having a pre-existing health condition was associated with increased odds of perceiving COVID-19 as a serious health risk among PEH. They also noted that men in their study had significantly lower odds of perceiving the virus as a serious health risk than women [85].

PEH’s perceived threat of COVID-19 impacted some aspects of their health and behaviour. In one study, increased loneliness was associated with a high self-perceived risk of contracting COVID-19 [84]. Kuhn et al. [87] reported PEH in their study who perceived the virus as a high threat were significantly less likely to be vaccine hesitant. Perceiving COVID-19 as a serious threat was also associated with increased odds of handwashing and social distancing among PEH in a study by Henwood et al. [85]. Finnigan [86] found that 27% of PEH surveyed in Sacramento, California, reported avoiding shelters due to fear of the virus. One study in Hamburg reported increased physician visits or likelihood of hospitalisation was not associated with fear of COVID-19 among PEH [82].

### 3.7. COVID-19 Vaccine Acceptance

Five studies measured attitudes toward COVID-19 vaccination. Three studies took place during December 2020 or later, when vaccines first became available in the U.S. and Italy [59,87,90], while two were conducted earlier in 2020, before vaccines received any emergency use authorisation [83,91]. Four studies examined attitudes exclusively among PEH [59,83,87,91]; vaccine hesitancy was reported as 41% [83] and 48% [87] in two studies, while vaccine acceptance was found to be 56% [91] and 64% [59] in another two studies. Fear of side effects (37%), wanting more information (30%), or rejecting all vaccines (37%) were cited as reasons for vaccine hesitancy among PEH in a study by Kuhn et al. [87]. Moore et al. [90] reported that housing insecurity was associated with more than sevenfold increased odds of vaccine resistance among American Americans living in the southern U.S.

Several factors that may contribute to vaccine hesitancy among PEH were identified. In France, Longchamps et al. [83] found increased odds of vaccine hesitancy among females (vs. males) and those living with a partner (vs. living alone) and decreased odds of vaccine hesitancy among those with no legal residence (vs. French/legal residence) and those with higher health literacy (vs. low). Similarly, Iacoella et al. [59] found that vaccine acceptance was higher among male PEH in Rome than females (74% vs. 59%, respectively). Kuhn et al. [87] reported that those trusting official sources were significantly less likely to be hesitant, and those engaging in highly protective behaviour were significantly more likely. A study in Los Angeles reported no significant differences regarding vaccine attitudes or uptake based on race/ethnicity, gender identity, sexual orientation, or testing history among young PEH (18–26 years old) [91]. Nearly 80% of participants in the same study felt that having access to primary prevention services and personal protective equipment (PPE) were important to promoting uptake, and 70% expressed that access to COVID-19 treatments, text-based prevention information and support, and the ability to get vaccinated in non-traditional medical settings were crucial for them to be vaccinated [91].

### 3.8. Housing

Many included studies reported substantial impacts of the pandemic on housing for PEH. Three studies indicated that the pandemic may have led to an increase in persons experiencing homelessness [21,86,92]. Between February and May 2020, Irwin et al. [21] reported a 91% increase in persons experiencing unsheltered homelessness in Arlington County, Virginia and noted an 88% increase in Black individuals and 48% increase in white individuals. Several studies also noted that the pandemic led some people to experience homelessness or be recognised by homeless support services for the first time [13,23,93]. Barbu et al. [94] reported that some persons newly experiencing homelessness during the pandemic had difficulties accessing emergency accommodation.

Several studies described how the pandemic exacerbated insecure living conditions for PEH. Four studies reported shelters restricting new admissions [12,46,95,96], and three reported shelters closing during the pandemic [44,73,94]. One study reported that authorities in France dismantled a squat [39], and in Salamanca, Spain, PEH were not allowed to live on the street during the initial lockdown period [26]. Additionally, Allaria et al. [13] reported 42% of PEH in their study changed accommodation since the onset of the pandemic. Court proceedings delayed due to the pandemic slowed intake of some PEH, particularly those leaving incarceration, into shelters, according to a study by Pixley et al. [97]. A study in Los Angeles by Tucker et al. [88] reported that 29% of young PEH (18–25 years old) indicated that the pandemic made finding a safe place to spend the night more difficult, and 42% indicated that it was now harder to find or keep stable housing. Some shelter residents in a study by Parkes et al. [23] also felt that support within accommodations was reduced during the pandemic.

Despite the negative impact of the pandemic on housing insecurity, many studies described instances in which housing supports were provided or improved for PEH. Ten articles noted that shelter services were expanded, or new temporary shelters were established, as part of the pandemic response [26,36,48,54,74,94,98,99,100,101], and 21 reported that PEH were temporarily housed in repurposed hotels [7,12,23,38,39,47,48,54,58,60,61,94,95,97,98,99,100,101,102,103,104]. Two studies mentioned participants staying in hotels but did not indicate whether their stay was related to temporary pandemic housing programmes [87,105]. Two studies reported that those leaving non-congregate hotel accommodation were given support in finding permanent housing [58,99]; 83% of participants in the study by Aitken [99] found suitable alternative accommodation. An additional four studies reported those leaving medical care sites were supported with a discharge plan for housing [72,99,106,107]. However, a report by Barbu et al. [94] noted that when some temporary accommodation supports eventually closed, some PEH to returned to rough sleeping. In Ireland, some PEH in temporary emergency accommodations expressed concern over uncertainties of future accommodations [101]. Three studies indicated changes made to reduce crowding provided increased privacy in shelters [44,96] or non-congregate settings [97]. Leonardi and Stefani [96] also noted that shelters in Turin, Italy began operating 24 hours a day, which fostered a sense of community among residents. Two studies reported shelters using incentives such as free meals, cigarettes, TV, and religious or spiritual events to keep clients indoors during the pandemic and reduce exposure to SARS-CoV-2 [67,71]. Both reported no positive cases among residents [67,71].

Isolation or quarantine accommodation for PEH who were symptomatic, confirmed cases was noted in 44 articles. Of these, the majority reported that access to temporary facilities to safely isolate or quarantine was provided [34,35,36,39,40,42,43,45,47,49,50,51,54,55,56,69,72,74,75,80,94,97,98,99,101,107], with 15 specifically describing hotels converted for this purpose [12,21,33,40,41,46,53,58,73,86,99,100,106,108,109]. Fuchs et al. [53] found that premature discontinuation of hotel isolation or quarantine was associated with experiencing unsheltered homelessness and requiring quarantine as a close contact. Wang et al. [107] described the implementation of a trauma-informed care site in Chicago, with high satisfaction reported among patients. Three studies mentioned isolation of individuals who were symptomatic or positive but did not provide further details as to where isolation occurred [48,68,110]. Six studies mentioned that PEH were unable to safely isolate or quarantine [25,39,69,85,93,96], three of which described a lack of safe isolation services in March and April 2020 [25,69,96]. Two studies noted that PEH were unable to quarantine or isolate in place if needed because of a lack of necessities such as food [85,93], hygiene, or medication [85].

### 3.9. Access to Personal Hygiene and PPE

Multiple studies discussed personal hygiene. Unmet need for showers [12,39], bathrooms [97], and other hygiene products or services [44,97] for PEH during the pandemic were reported by several studies. In a study by Riley et al., [111] 66% of women experiencing homelessness or unstable housing reported one or more subsistence needs, defined as insufficient access to food, clothing, housing, or hygiene resources. In some cases, barriers to hygiene were exacerbated by the pandemic; reduced access to showers [88,94,109], toilets [94], laundry [88,94], and other personal hygiene products and services [13] were discussed by several studies, with some noting that barriers were due, at least in part, to public closures [12,94,97,109]. However, some studies indicated that PEH had access to showers [39,40,55,68,94,100], toilets [39,68,94,100], or laundry [94,100]. Access to general hygiene products or services was also noted in five studies [48,53,86,100,101], all of which were provided by organisations or shelters. In a few studies, access to showers [39,55] and hygiene products [11] was reported as improving for some PEH during the pandemic.

Other personal hygiene concerns brought up by studies included cleaning supplies and sharing of substances. One study reported that PEH had access to cleaning supplies [60], three reported unmet need [13,40,94], and one indicated decreased access as a result of the pandemic [11]. Sharing of substances among PEH as an infection risk was brought up by three studies [12,88,110]. One study noted that cigarettes were commonly shared among PEH [12], and one found that many PEH avoided sharing cigarettes or drugs because of the pandemic [88]. In addition, Steer et al. [110] reported positive outcomes of a disposable cup intervention to reduce drink sharing among PEH, particularly among those using alcohol.

Hand hygiene among PEH was discussed by numerous studies. Many noted that PEH were able to wash their hands [11,13,41,42,44,47,66,85,88,100], access soap [11,42,47,88], or access hand sanitiser [11,39,40,41,60,69,88,100]. Use [100] or provision [66] of gloves to PEH was also mentioned by two studies. However, unmet needs for handwashing facilities [11,12,88], soap [11,88], and hand sanitiser [11,40,55,88,94] were also reported. Montgomery et al. [11] reported that public closures and price surges of hand hygiene supplies triggered by the pandemic reduced access to soap, hand sanitiser, and handwashing facilities for PEH in Atlanta, Georgia, though supplies and handwashing stations were later provided. The same study found that unsheltered PEH were more likely to rely on hand sanitiser, bottled water, and disinfecting wipes for hand hygiene, and some PEH in shelters expressed concerns about crowding and long lines to wash hands [11]. Henwood et al. [85] reported that among PEH living in Skid Row, those living in single room occupancies with shared bathrooms and those with mental health conditions were nearly half as likely to report hand washing compared to those in studios.

Personal protective equipment (PPE) was discussed in several studies. PEH were reported as having access to facemasks or face coverings in 21 studies [12,13,23,33,39,41,46,47,48,52,54,55,58,60,66,69,74,88,93,98,100], among which nine reported that these were required to be worn in shelters [12,33,41,46,48,52,54,100] or quarantine and isolation sites [74]. Three studies noted that PPE generally was available to PEH [25,58,100]. PPE [26] and masks [86] were also reported as being worn during interviews for two studies. Additional studies noted that facemasks were enforced during health check-ups [20] or in public [42], and encouraged while awaiting test results and in general patient areas of a health facility [51]. Rodriguez et al. [12] described difficulties enforcing masks among PEH in shelters, with some giving up and only requiring them to be worn by staff. Unmet need for facemasks/face coverings [40,55,88,94] or PPE [23,69,97] for PEH was discussed in seven studies.

Physical distancing was also noted in several included articles. PEH were reportedly able to follow social distancing guidelines in 14 studies [13,33,39,40,41,42,44,46,47,52,74,88,93,98,100], among which seven reported that distancing was enforced in shelters [40,41,44,52,98,100] or quarantine and isolation sites [74]. Eight studies reported that some PEH were unable to follow social distancing guidelines [12,25,33,39,46,88,94,103], sometimes even despite markers indicating recommended spacing [25,103]. Reluctance to follow and ambivalence regarding the importance of social distancing among PEH was noted in four studies [12,23,25,93]. Physical distancing was enforced among some PEH accessing homeless support services in the U.S. [97], during health check-ups in Salamanca [20], and during interviews for a study in Spain [26] and encouraged among patients accessing a psychiatric emergency room in Los Angeles [51] and shelter residents in Washington State [55]. One study in Ireland reported accommodation services were expanded to support social distancing [101]. Henwood et al. [85] reported that among PEH living in Skid Row, those living in single room occupancies and those with mental health conditions were nearly half as likely to report consistent social distancing than those in studios. In a study by Kuhn et al. [87], 42% of PEH reported high COVID-19 protective behaviour, measured by frequency of wearing a mask, washing hands, distancing from others, and avoiding touching their face.

### 3.10. Access to Food

There were mixed impacts of the COVID-19 pandemic on access to food for PEH. Three studies reported unmet need for food during the pandemic [85,97,111]. Riley et al. [111] measured subsistence needs, defined as insufficient access to food, clothing, housing, or hygiene resources, and found that 66% of women experiencing homelessness in the study had at least one unmet subsistence need. Additionally, when asked if they would be able to shelter in place for 14 days if needed, 45% of PEH in a study in Los Angeles responded no, with 91% citing lack of food as a reason [85]. Three studies reported finances as a barrier to PEH accessing food [88,93,94]. In addition, five studies reported that food services for PEH were reduced or halted because of COVID-19 [12,25,97,102,109]; three reported that these were closed or limited because of risk of virus transmission among PEH accessing services [12,25,102], and two did not provide specific reasons for closure [97,109]. Several studies noted that COVID-19 reduced access to food for PEH [12,13,88,94,112]. In a study by Tucker et al. [88], 54% of PEH indicated that the pandemic made it harder to get enough food to eat. In a large city in France, when compared with PEH in shelters, persons sleeping rough were significantly more likely to have difficulty accessing food (24% vs. 60%, respectively) and water (5% vs. 39%, respectively) as a result of the pandemic [13]. The same study noted that access to food assistance was especially reduced for those more recently homeless compared with those living rough or in slums longer, who had established networks [13]. In Scotland, restrictions on movements limited options for some PEH, who were no longer able to travel to places with cheaper food [112]. Transportation barriers were also noted in a study by Gaeta et al. [74].

Some articles did report examples of food needs being met, or even improving, as a consequence of the pandemic. Two studies reported outreach services were able to meet food needs for PEH [60,109], and one study noted that several participants felt they had easier and more regular access to food due to the support received during the pandemic [99]. It was also frequently noted that meals were provided to individuals as patients [41,58,72,74,101] or residents of temporary accommodation [25,26,33,46,53,58,67,71,94,96,98,99,100,101,102,112]. However, a report from Scotland explained that while asylum seekers were rehoused into hotels and provided meals, the food was often poor in nutrition or culturally inappropriate, leading to malnourishment and mental health issues [112]. In Ireland, PEH in temporary emergency accommodation felt that lack of cooking facilities was a barrier to eating proper meals, and several suggested improved quality and frequency of meals [101].

### 3.11. Substance Use

Active substance use among PEH during the pandemic was noted in 36 studies [8,11,12,13,20,23,25,26,38,47,51,61,62,68,72,76,81,82,84,88,89,92,94,96,99,101,102,103,104,107,108,110,113,114,115]. Substances used included alcohol [8,13,23,25,26,47,61,62,68,72,82,84,88,89,94,99,101,102,108,110,115], tobacco [8,12,13,38,61,62,68,88,108], cannabis [23,26,51,62,88,108], cocaine [26,51,62,72,113], methamphetamines [72,104,114], heroin [62,99,113], stimulants [108,110,113], unprescribed benzodiazepines [23,72,113], fentanyl [104,113], amphetamines [51], and gamma-hydroxybutyrate [108]. An additional four studies noted general opioid use [26,72,108,113], and 17 reported current substance use without further specification [11,12,13,20,25,38,47,61,68,76,92,96,101,102,103,107,110,114]. Increased use of substances was reported in four studies [20,23,88,104]. Increased use of marijuana (28%), tobacco (20%), and alcohol (20%) during the pandemic was reported by some young PEH (18–25 years old) in Los Angeles [88]. In Scotland, the emotional impact of lockdown, isolation, and reduced support services contributed to increased drug use among some PEH [23]. Similarly, individuals in a study by Scallan et al. [104] reported increasing substance use following loss of housing supports. Aguilar et al. [20] reported an increase in relapses among PEH during the first ten weeks of confinement in Spain, though this finding was reported in the discussion only, and supporting data was not available. One study reported reduced drug use among some PEH was facilitated by increased privacy and sense of safety they experienced since shielding or self-isolating in emergency accommodations [101].

Many included studies discussed access to substance use treatment for PEH during the pandemic. Treatment was reported as available in 22 studies [8,12,20,23,25,26,51,58,65,72,85,88,93,97,99,101,102,105,107,108,113], and access or uptake was noted as improving in 12 studies [23,25,26,72,88,97,99,102,104,107,108,113]. Tucker et al. [88] reported that 13% of young PEH (18–25 years old) in their study found substance use services easier to access since the onset of the pandemic. Nine studies noted that access to treatment improved within the context of accommodation services [23,26,58,72,99,101,104,107,108], five of which specifically noted that PEH initiated treatment for the first time within the service [23,72,99,101,108]. Participants in a study by Pixley et al. [97] reported that PEH were more open and accepting to substance use treatment following the rollout of alternative services and improved housing standards in noncongregate sheltering. Both Fitzpatrick et al. [102] and Parkes et al. [23] reported that reduced financial support from the public led PEH to seek prescription medication rather than illicit substances. The use of telehealth to support treatment was noted in seven studies [20,25,65,101,107,108,113]. Preventative measures implemented for fatal overdoses among PEH were reported in two studies, with both reporting no fatal overdoses at the time of the studies [72,108].

Five studies discussed reduced access to substance use treatment services during the pandemic [12,23,25,88,101], three of which specifically noted that services for alcohol use disorders were limited [23,25,101]. Tucker et al. [88] reported that 32% of participants in their study reported that accessing substance treatment services was harder since the onset of the pandemic. Service providers interviewed by Rodriguez et al. [12] also expressed concerns regarding PEHs’ access to substance treatment, with some noting that reduced addiction treatment services led some individuals to relapse. Another study in California by Appa et al. [92] reported that fatal overdoses increased among PEH in the eight months following the onset of the pandemic (defined as 17 March 2020) compared with the eight months prior.

### 3.12. Mental Health

Numerous studies discussed the impact the pandemic has had on PEH’s mental health. Poor mental health was reported in five studies [51,84,101,111,115]. A survey of PEH in Hamburg revealed that 32% of PEH had problems with anxiety or depression and that those with health insurance had lower odds of experiencing these conditions [115]. Another study using the same dataset from Hamburg reported that 49% of those surveyed felt lonely, with increased loneliness associated with male gender, being single, originating from Germany, high frequency of sharing a sleeping space with more than three people, and a higher self-perceived risk of contracting COVID-19 [84]. In San Francisco, 55% of women experiencing homelessness and unstable housing had depression, and 42% had anxiety; factors significantly associated with depression and anxiety included recent homelessness, unmet subsistence needs, and social isolation [111]. Increased difficulties accessing care for chronic medical conditions also increased risk of screening positive for anxiety more than threefold, and for depression, sixfold [111]. Cardenas et al. [51] reported that the majority of individuals presenting to a psychiatric emergency room in Los Angeles were PEH.

Nine studies indicated that the pandemic led to poorer mental health outcomes among PEH [12,20,23,25,88,94,96,97,101]. Disruptions to routines [12], feelings of loneliness [88,94,96,101], exclusion, confinement, [94] nervousness, [94,101] hopelessness [88], exacerbation of pre-existing mental health problems [23], and reduced access to services and counselling [12,101] were all seen as contributing to negative mental health during the pandemic. In temporary emergency accommodations in Ireland, 39% of PEH surveyed in May and June 2020 reported worse mental health than one year prior, and 21% self-harmed, attempted suicide, or had suicidal thoughts in the past month [101]. Aguilar et al. [20] reported increased psychological destabilisation among PEH during the first ten weeks of confinement in Spain, though supporting data for this finding were not available. Three studies noted mental health improving for some PEH during the pandemic [96,101,116]. In Turin, Leonardi and Stefani [96] described how night shelters that shifted to 24/7 services became residential communities, providing stability and improving mental health for some residents. A study by the Irish Health Service Executive (HSE) [101] reported that 39% of PEH surveyed in emergency accommodations self-reported improved mental health compared with one year prior; increased privacy, sense of safety, and rebuilding relationships with family since shielding or self-isolating was noted by some as contributing to improved mental health. Additionally, a positive association was found in changes in moderate or total physical activity and mental well-being and self-esteem among young PEH (16–24 years old) between the four weeks before and after the initial lockdown in the UK in a study by Thomas et al. [116].

Access to mental health services was addressed in 11 studies [23,25,26,58,67,72,88,97,99,101,106]. Of these, five indicated that mental health support was provided within temporary accommodations [58,67,72,101,106], and four indicated that PEH’s access to support improved during the pandemic [25,26,99,101]. Parkes et al. [25] reported that telephone and online support groups helped PEH in Scotland to maintain and improve their mental health. In a temporary shelter in Spain, significantly more patients were prescribed psychotropic drugs by the end of the programme than at the beginning (82% vs. 59%, respectively) [26]. Aitken [99] reported that hotel accommodation provided a safe space for PEH experiencing mental health conditions and increased willingness to engage with support. Two studies reported that some PEH were discharged to mental health programmes after leaving temporary accommodations [26,72]. Unmet need for mental health services was noted in five studies [13,23,88,97,101]. In France, 24% of PEH in a study by Allaria et al. [13] reported unmet mental health needs, with the highest unmet need reported among those sleeping rough (33%), and the lowest, among those living in squats (17%). Parkes et al. [23] reported that telephone and online support groups were not enough to offset increased social isolation due to the pandemic. Similarly, Pixley et al. [97] noted barriers to accessing online support among PEH. In Ireland, some PEH in temporary accommodations noted that they were unable to access their psychiatrists and that provided mental health services could be improved [101]. A study in Los Angeles by Tucker et al. [88] reported that 44% of young PEH (18–25 years old) felt that accessing mental health counselling was more difficult since the onset of the pandemic.

### 3.13. Access to Health Services

Beyond healthcare directly related to COVID-19, substance use, or mental health, several studies discussed PEH access to healthcare during the pandemic. Seven articles reported unmet health needs [12,13,65,101,115,117,118]. In Marseille, France, 17% of PEH reported unmet physical health needs, with the highest unmet need reported among those in shelters (21%), followed by those living rough (18%) and in squats (12%) [13]. van Rüth et al. found that only 69% of PEH living in Hamburg during the pandemic reported having health insurance [115]. In Ireland, 32% of surveyed PEH in temporary emergency accommodations did not have an up-to-date care plan, and 15% did not know [101]. Substantially fewer PEH in county Dublin (35%) reported having an up-to-date care plan than in Galway, Limerick, Clare, and Tipperary (70%) [101]. For some, the large shift toward telemedicine was a barrier to health services. In Indiana, PEH were unable to avail of telehealth services because of lack of access to phones, computers, or places to charge or store devices [12]. Another study found that U.S. veterans experiencing homelessness were 11% less likely to use video care during the pandemic than those not experiencing homelessness [117]. Similarly, Hickey et al. [65] reported that PEH accounted for 9% of those reached prior to a scheduled telehealth visit and 17% of those not reached. At a large HIV clinic in San Francisco, Spinelli et al. [118] reported that PEH were offered telehealth visits significantly less often than the average population (32% vs. 54%, respectively) and had fewer no-shows during shelter-in-place (1–30 April 2020) than the average population pre-shelter-in-place (1 December 2019–29 February 2020), and that viral non-suppression was higher among PEH during the pandemic than before. Barriers to primary healthcare were also exacerbated by the pandemic, with PEH in Edinburgh being turned away from the A&E for problems unrelated to COVID-19 and unable to meet with general practitioners (GPs) or access wound care, sexual health, or dentistry services [23]. Reduced access to STD services such as condoms, testing, or PrEP due to the COVID-19 pandemic was also reported in a Los Angeles-based study [88].

While some articles described unmet healthcare needs among some PEH, eight studies reported instances in which health needs of PEH were supported during the pandemic [25,60,71,99,100,107,109,114]. Three studies described medication delivery for PEH; two described this in relation to enabling PEH to shield [25,71], while the third delivered prescriptions to PEH who were isolating with COVID-19 [107]. In addition to prescription delivery, PEH isolating in care sites in Chicago were supported through telehealth visits and transportation to and from outpatient haemodialysis [107], and those residing in three shelters on the Slovakia borders received regular GP visits [71]. Some patients admitted to an Intermediary Care Unit in Edinburgh for recovery from acute illnesses were able to reengage with primary care, access hepatitis-C treatment, or receive care for chronic health conditions [99]. One HIV clinic in San Francisco reported that the proportion of PEH visiting the clinic each month was similar before and during the pandemic and that viral suppression did not worsen among patients [114]. In addition, 15% of the patients were temporarily housed in hotels, enabling navigators to conduct both phone and in-person outreach [114]. An intervention for persons living with HIV with experience of homelessness in Boston provided phones to patients without devices, facilitating biweekly contact to ensure that medical and prescription needs were met [60]. The intervention found that 57% of patients that were unhoused kept their appointments with their HIV primary care providers, though this was a significantly lower proportion than among those who were currently housed (75%) [60]. Redondo-Sama et al. [109] described collaboration between social workers and health services, enabling advocacy for vulnerable patients, and Brown and Edwards [100] reported that health support was delivered to unsheltered homeless encampments in California by the Emergency Operation Centre, though details of the health support were not provided.

In some cases, the pandemic improved access to health services for PEH. Some residents of temporary emergency accommodation in Ireland reported accessing new supports, such as primary care services, on-site nursing, housing support, and project worker support [101]. The same study noted that the number of respondents engaging with keyworkers or case managers increased during the outbreak period (April–June 2020) compared with before (September 2019–March 2020) [101]. In England, homeless services’ closer connections with the health services helped PEH to receive better health assessments [102]. In addition, some providers of HCV test and treat interventions for those temporarily housed in England reported that increased freedom and flexibility allowed them to provide clients all of their medication upon treatment initiation, reducing the consequences of losing contact while people moved between accommodations [103]. Providers also felt that the lockdown and accommodations provided time and space for some PEH to reflect on and reengage with their health [103]. A study by Cironi et al. [95] in New Orleans reported that 60% of those testing positive for HCV as part of a pilot program among persons in COVID-19 temporary housing were previously unaware of their infection. The program was therefore able to communicate diagnoses with residents and link them with follow-up care [95].

### 3.14. Other Health Impacts

A few studies explored other health impacts of the pandemic, such as violence [97,105], sense of safety [101], physical activity [101,116], emergency department use [101,105], local health centre use, quality of life, and general health status [101]. Pixley et al. [97] explained that while domestic and interpersonal violence had been associated with homelessness and housing insecurity prior to the pandemic, new financial fears or fear of SARS-CoV-2 may prevent some individuals from leaving violent or abusive situations. A study by Riley et al. [105] noted that 33% of women experiencing homelessness or housing insecurity in San Francisco decided where to sleep based on avoiding violence during the pandemic. In Ireland, 70% of surveyed PEH in temporary emergency accommodation reported feeling safe or very safe in May and June 2020, and 46% reported feeling safer than they did one year prior [101]. Regarding physical activity, some participants in the same study in Ireland noted more appreciation for exercise in the emergency accommodations while others felt that not having a gym or facing difficulties walking negatively impacted their well-being [101]. Thomas et al. [116] found that physical activity among young PEH (16–24 years old) generally increased in the UK during the four weeks following lockdown restrictions introduced in March 2020 compared with four weeks before. Increased physical activity following lockdown was significantly higher in participants considered ‘inactive’ prior to lockdown than in those considered ‘active’ [116]. Riley et al. [105] reported that, unlike the general population, women experiencing homelessness and housing insecurity in San Francisco did not reduce emergency department use during the pandemic. Experiencing homelessness was significantly associated with emergency department use [105]. In Ireland, visits to local health centres and the emergency department declined among PEH who were accessing emergency accommodations in most areas during the outbreak period (April–June 2020), except county Galway, where a 17% increase in emergency department use was reported [101]. In the same study, 54% of respondents described their quality of life as good, very good, or excellent; 28% described it as fair; and 18% self-reported poor or very poor quality of life [101]. Respondents also self-reported their general health status; 46% described it as better than one year prior, 34% reported it as worse, and 30% indicated no change [101].

## 4. Discussion

This review aimed to provide an overview of the impacts of the COVID-19 pandemic on the health and well-being of PEH in North America and Europe. Literature on the topic was broad in terms of study design, measures, and outcomes across a number of countries. Impacts on health and well-being that were identified included SARS-CoV-2 infection, morbidity, mortality, and hospitalisation, fear of SARS-CoV-2 infection, access to housing, hygiene, PPE, food, as well as mental health, substance use, and various other health-related treatment services.

The findings from this review indicated that PEH are at high risk for infection with COVID-19. This was expected, given that many PEH live in congregate settings such as shelters or encampments that facilitate virus circulation [6,9]. High transmission of other pathogens has been previously documented in homeless shelters [77,119,120]. While it was beyond the scope of this review to conduct a meta-analysis of SARS-CoV-2 prevalence among PEH, Mohenspour et al. [28] suggest a baseline prevalence of 2.1% among PEH in homeless shelters, increasing to 29.5% during outbreaks.

This review found that PEH may be at high risk for hospitalisation and, in some cases, experience higher mortality from COVID-19 than their counterparts in the general population. These findings were not surprising, given that PEH are generally at higher risk for comorbidities and poorer health conditions that may contribute to worse outcomes following SARS-CoV-2 infection [6,10] and that PEH ages 50 or older have been found to have more geriatric conditions than stably housed individuals 20 years older [121]. Given this increased risk, the lower CFR among PEH ages 65 and older than among their counterparts in the general population reported by Leifheit et al. [5] was unexpected. However, it may be the case that lower life expectancy among PEH [119,122] led to fewer individuals ages 80 or older than in the general population, or perhaps survivor bias was a factor, with those living to age 65 unsheltered being less frail than those in the general population [5].

Given the increased risk for infection, morbidity, and mortality, housing support for PEH has been a priority in the pandemic response across countries in North America and Europe [22,123,124,125] and was a key theme among studies in this review. The urgency to rapidly rehouse PEH during the pandemic has largely been in an effort to reduce virus transmission in congregate settings and shield individuals who are at increased risk for severe outcomes of SARS-CoV-2 infections. However, previous studies have indicated that benefits of housing support may extend far beyond outbreak and infection mitigation. Indeed, improvements in mental health [126,127], quality of life [128,129], substance use [130,131], and reduced criminal justice system involvement [128,132] and hospital use [133,134] have been reported following housing support provision for PEH. During the pandemic, many temporary housing centres connected individuals to various services accessible within their accommodation, providing a sense of stability, independence, and support that enabled some PEH to focus on their health, relationships, and work toward permanent housing.

This review found that the pandemic had a substantial impact on mental health, substance use, and day-to-day health among PEH. Feelings of isolation, barriers and disruptions to support, and economic hardship due to the pandemic have contributed to poorer mental health among the wider population [135,136,137] and have likely been magnified for PEH, who already experience higher rates of mental health conditions [138]. These factors, as well as depression and anxiety, have also been associated with changes in substance use during the pandemic [139,140,141], findings that were supported by this review. In an effort to minimise risk of virus transmission, many health services shifted toward service provision via telehealth [142,143,144]. Barriers to information technology, such as poor access, affordability, and charging devices, have been previously documented among PEH [145] and may hinder their use of telemedicine. These barriers may have contributed to the lower use of telehealth services among PEH that was reported by many articles in this review, particularly concerning access to health services beyond mental health and substance use.

While this review found that the pandemic exacerbated many barriers to health and support services for PEH, it also provided unique opportunities to improve access and enhance services for some PEH. Proactive outreach to PEH concerning substance use issues has been shown to improve linkages to treatment and prevent drug-induced deaths [146,147,148], findings that were supported by studies in this review. Additionally, several studies have found that on-site service delivery within accommodations facilitates access and may lead to better outcomes for PEH [149,150,151,152]. The provision of services and support via telehealth and within accommodations not only enabled some PEH to access support within a safe and secure environment but likely reduced the barriers PEH face. Indeed, a recent study by Barile et al. [153] reported that many PEH experienced difficulties accessing services due to transportation barriers, which may be why providing phones and outreach to PEH in places of residence brought about positive impacts in some of studies included in this review.

Provision of or linkage to health services within accommodation or other support networks was highlighted as valuable by several included studies and enabled some PEH to access treatment for the first time. On-site service delivery allowed PEH to address their needs in a safe environment and has previously been shown to facilitate healthcare access [149,150,151]. However, provision of mental support services to persons sleeping rough was explored less often by studies in this review. Given that unmet mental health needs were highest among this population in a study by Allaria et al. [13], facilitators of service access and provision should be further explored. Additionally, among studies that addressed substance use support, there was an overwhelming focus on opioids or other illicit substances. Very few examined services available for those with alcohol addiction, and two studies noted that these services were limited [23,25]. Given the time-sensitive nature of a rapidly evolving pandemic, and given that opioid use disorder (OUD) is associated with higher risk for fatal overdoses than alcohol use disorder (AUD) [154], the prioritisation of resources for OUD is understandable. However, studies have indicated that AUD is common among PEH and often cooccurs with other substance use disorders [155,156]. It is therefore important for future studies to explore the impact of the pandemic on PEH with AUD, as well as barriers to and facilitators of support services.

Studies in this review indicated that the pandemic had a significant impact on access to personal hygiene for PEH, and compliance with public health guidance varied across the population. Paudyal et al. [157] suggested lack of resources, multimorbidity, low health literacy, and social influences as reasons for low self-care among PEH. Given that many PEH rely on public restrooms [158,159], the negative hygiene consequences of widespread closures of nonessential public businesses to reduce COVID-19 transmission [160,161] was expected. Difficulties complying with social distancing and public health guidance were also unsurprising, given the often-crowded living conditions of shelters and encampments [162,163]. Globally, studies have reported a positive relationship between fear of COVID-19 and compliance with public health measures [164,165,166,167]. Studies in this review supported this relationship and may explain why some PEH reported stricter adherence to guidelines than others. Additionally, previous research has suggested that access to private showers, restrooms, and running water within day centres for PEH promote a greater sense of security and safety [168,169]. Therefore, increasing availability and access to private restrooms and hygienic resources could have twofold benefits: increased engagement with personal hygiene and compliance with COVID-19 guidelines [11,40,88,170,171].

PEH have been a prioritised group in COVID-19 vaccine rollouts in Europe and North America because of their increased risk for transmission, severe disease, and death from the virus [172,173]. However, several articles in this review indicated hesitancy among PEH toward COVID-19 vaccinations. Previous studies have shown increased vaccine hesitancy [174] and lower uptake among PEH with other vaccines than among the general population [175]. Documented barriers to uptake among PEH include poor access, lack of information, low perceived importance, and mistrust in vaccines [174,175,176]. This may explain why young PEH in a study by Hsu et al. [80] felt that access to vaccines in non-traditional settings and text-based information were critical to support COVID-19 vaccination in the population. Similar findings were reported by Doroshenko et al. [177] regarding other vaccines, and several studies have demonstrated increased uptake among PEH in response to direct outreach and education programmes [178,179]. As vaccines and boosters continue to roll out, more studies should examine how information and access is provided to PEH in order to better inform and facilitate future vaccination campaigns.

Food insecurity has long been reported among PEH [180,181,182,183,184] and the pandemic has exacerbated this for many PEH and other vulnerable populations [185,186,187,188]. Drivers of food insecurity throughout the pandemic have included increased financial insecurities, restrictions on movements, disruptions to food distribution services, and supply shortages [189,190,191]. While food insecurity was frequently reported among studies in this review, many studies also noted the rapid response of governments and organisations providing food for PEH through accommodations, food banks, and delivery of food parcels. Importantly, however, risks of food insecurity may be magnified among PEH sleeping rough or those with fewer connections to support services, as was noted by Allaria et al. [13]. Housing support services have been shown to reduce food insecurity [184,192] and may be especially important in combating hunger when food support services are closed or limited to curb the spread of an infectious disease.

Finally, very few studies examined how violence impacted PEH during the pandemic. Several studies have indicated that domestic violence increased among the general population following COVID-19 related lockdowns [193,194]. ‘Stay at home’ orders have exacerbated unsafe living conditions for those in abusive environments and impacted many survivors’ access to help and coping mechanisms [195,196]. As suggested by Pixley et al. [97], fears and uncertainties during the pandemic have also likely led some PEH to remain in unsuitable and unsafe environments. Given that violence is one of the leading causes of housing instability and homelessness [197,198,199,200], more research exploring the impact of violence during the pandemic on homelessness, as well as how resources for PEH fleeing violence have been affected, is critical to inform best practices for policy and support services.

### Limitations

Although the present review provided an extensive overview of the health impact of the COVID-19 pandemic on PEH in North America and Europe, some limitations must be noted. First, limiting inclusion to only studies written in English may have excluded relevant papers from more diverse contexts. Additionally, because of time constraints, only 16 articles were double extracted by two reviewers. While studies that were extracted by only one reviewer were cross-checked for validity by a second reviewer, some relevant data may have been missed. Finally, given the breadth and heterogeneity of literature on this topic, it was beyond the scope of this review to conduct a quality analysis or interpret findings within identified subthemes. Future studies should examine these subthemes in more depth and compare health impacts across countries.

## 5. Conclusions

The review identified a number of health impacts of the COVID-19 pandemic on PEH in North America and Europe, such as SARS-CoV-2 infection, morbidity, mortality, and hospitalisation, fear of SARS-CoV-2 infection, access to housing, hygiene, PPE, and food, as well as mental health, substance use, other health-related outcomes, and treatment services. However, some gaps in the literature were identified. The majority of studies addressing substance use among PEH focused on opioids and other illicit substances, with few focusing on persons using alcohol. Future studies should examine the impact of the pandemic on PEH experiencing alcohol addiction and barriers to and facilitators of support. Secondly, few studies examined mental health support and services for persons sleeping rough during the pandemic, a population that may be especially vulnerable. Future research exploring barriers and facilitators to mental health support for those sleeping rough may help inform interventions and improve access. Additionally, more research exploring the impact of the pandemic on interpersonal violence against PEH is needed. As the pandemic continues, and as the world moves toward a post-pandemic normal, studies should explore how supports provided during the pandemic for PEH have changed or evolved, and how lessons learned can be used to inform policies and practices for sustained and improved healthcare for PEH.

## Figures and Tables

**Figure 1 ijerph-19-03219-f001:**
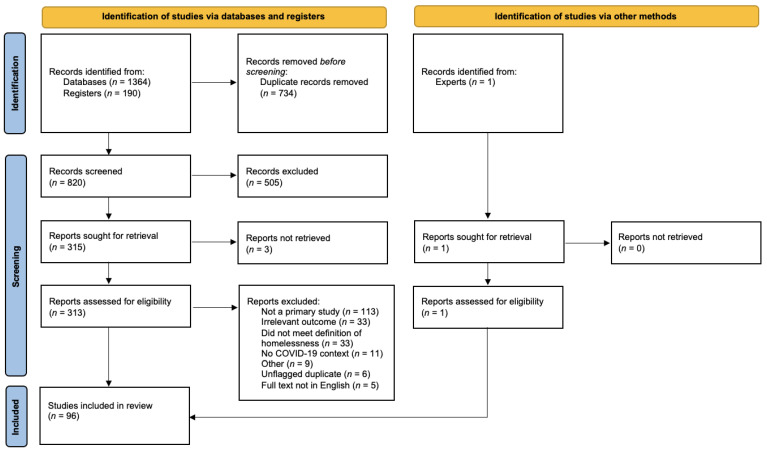
PRISMA Flow Diagram.

**Table 1 ijerph-19-03219-t001:** Study characteristics.

Country		Study Design	
U.S.	51	Cross-sectional	30
U.K.	9	Unspecified *	7
France	9	Longitudinal	4
Canada	6	Mixed methods	4
Spain	5	Pilot	4
Italy	4	Case study	4
Germany	4	Qualitative	3
Denmark	2	Report	3
Belgium	2	Case report	2
Multiple	2	Case series	2
Slovakia	1	Retrospective	2
Ireland	1	Cross-sectional seroprevalence	2
**Primary Population of Interest**		Analytical observational	1
PEH	74	Community-based participatory research	1
General patient populations	6	Cross-sectional community-based	1
People living with HIV	5	Cross-sectional community-based surveillance	1
Multiple vulnerable groups	3	Cross-sectional multicentre cohort	1
COVID-19 patients	3	Cross-sectional retrospective chart review	1
Persons using drugs	2	Descriptive	1
U.S. Veterans	1	Disease prevention protocol	1
African Americans in Southern U.S.	1	Interrupted time series	1
Criminal justice-involved women	1	Longitudinal ecological	1
**Publication Type**		Matched-case control observational	1
Journal article	48	Nationwide cross-sectional seroprevalence	1
Brief report	8	Nonconcurrent cohort	1
Research letter	6	Nonrandomised observational pre/post	1
Short communication	4	Nonrandomised pre/post	1
Case study	4	Observational retrospective	1
Preprint	4	Point prevalence	1
Report	4	Population-based prospective	1
Letter to the editor	3	Population-based retrospective e-cohort	1
Research note	2	Pragmatic randomised controlled trial	1
Weekly report	2	Prospective	1
Notes from the field	2	Qualitative exploratory	1
Rapid communication	1	Quality improvement program	1
Practice full report	1	Rapid case study	1
Case report	1	Repeated cross-sectional	1
Brief research report	1	Retrospective chart audit	1
Concise communication	1	Retrospective cohort	1
Short report	1	Retrospective cross-sectional	1
Research brief	1	Retrospective serological	1
Review	1		
Briefing report	1		

* Some studies did not specify the study design used. Appendix A provides more information as to the methods used for data collection.

## Data Availability

Not applicable.

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
