# Peer review of "A Scoping Review of the Health Impact of the COVID-19 Pandemic on Persons Experiencing Homelessness in North America and Europe"

_ijerph, 2022, doi:10.3390/ijerph19063219_

Round 1
Reviewer 1 Report
This is a well thought out article and will be of value to further studies around PEH. I only have a few comments:
- This sentence needs to be reviewed for grammar: While previous reviews have examined the health consequences of COVID-19 on PEH, they have largely focused on infection rates and control measures [27,28] the pandemic response [29,30].
- I would suggest adding in why you have chosen the specific criteria for the study selection that was used.
Author Response
This is a well thought out article and will be of value to further studies around PEH. I only have a few comments:
- This sentence needs to be reviewed for grammar: While previous reviews have examined the health consequences of COVID-19 on PEH, they have largely focused on infection rates and control measures [27,28] the pandemic response [29,30].
Thank you for highlighting this oversight. We have amended the sentence to read as follows, “While previous reviews have examined the health consequences of COVID-19 on PEH, they have largely focused on infection rates and control measures [27,28] or the pandemic response [29,30].”
- I would suggest adding in why you have chosen the specific criteria for the study selection that was used.
Thank you for this comment. We have added the following to expand on our exclusion criteria, “The decision to exclude studies that defined history of homelessness as homelessness pre-dating 1 January 2019 was agreed upon by the research team to ensure that included studies were based on up-to-date housing data for the respective population. Additionally, as the World Health Organization declared COVID-19 a pandemic in March 2020 [32], publication in or after this month was deemed by the research team to be an appropriate requirement for included literature, in order to capture the health impacts of the pandemic on PEH.”
Reviewer 2 Report
This is an excellent review, mapping evidence to explore the impacts of the COVID-19 pandemic on the health and well-being of PEH in North America and Europe, and identifying major gaps to support further research and practice development.
Author Response
This is an excellent review, mapping evidence to explore the impacts of the COVID-19 pandemic on the health and well-being of PEH in North America and Europe, and identifying major gaps to support further research and practice development.
Thank you for this feedback and for taking the time to consider our review for publication.
Reviewer 3 Report
The article named “A Scoping Review of the Health Impact of the COVID-19 Pandemic on Persons Experiencing Homelessness in North America and Europe” is proposed for publication after minor (format) revision.
It is proposed a scoping review to delve into Covid-19 disease and homelessness as a central focus in a specific framework: North America and Europe. The authors justify the need to study this issue because the homeless population is one of the most vulnerable group to Covid-19, not only because of the potential infection and physical health problems, but also because of the psychosocial effect on their life context and mental health impairment it has had. As noted in lines 42-55:
“For example, the large-scale closure of public buildings and facilities reduced access to toilets and basic hygiene and sanitation for those who were unsheltered [11,12]. Less financial support from the general public due to stay-at-home advisories and closure or re-organisation of food and social support services also contributed to difficulties accessing food assistance and hygiene products (…)”.
In the Methodology section, the authors correctly reflect how the systematic search was conducted across five databases (EMBASE, Medline, CINAHL, PsychINFO, Web of Science) and Google Scholar (grey literature).
I really appreciate how they describe in Appendix A the search strategy. However, it is suggested that the authors present it in a figure/table format, to make it more visually appealing for the audience.
After the combined effort of 3 researchers, it was decided to select the articles according to the following criteria (lines 89-90; PEH= Persons experiencing homelessness):
“1) included primary data related to the impact of COVID-19 on health or well-being of PEH; and 2) were written in English”.
Besides, it is properly stated the exclusion criteria (lines 91-101). I would like to emphasize that articles prior to 2020 were excluded, so that they were effectively carring out an examination of the most recent and current context in this group.
It is suggested though that, in future studies, the authors could include these previous articles in order to be able to make a comparison between their earlier and present life contexts after Covid-19. Also, another future line of research could be to compare the homeless population with other vulnerable populations or even the normalized population towards to explore differences and similarities in terms of the influence of Covid-19 on their health impact and psychosocial context. Perhaps even the differences between socioeconomically disparate contexts (countries) can be explored.
Figure 1 shows PRISMA Flow Diagram in a very visual way that makes it easy to read and understand the procedure the researchers have followed.
The results are correctly collected and described. Besides, they are presented in an appropriate organization (starting in line 119):
3.1. Screening Process
3.2. Overview of Included Studies
3.3. SARS-CoV-2 Infection & Morbidity
3.4. COVID-19 Related Hospitalisation
3.5. COVID-19 Related Mortality
3.6. Fear of COVID-19
3.7. COVID-19 Vaccine Acceptance
3.8. Housing
3.9. Access to Personal Hygiene and PPE
3.10. Access to Food
3.11. Substance Use
3.12. Mental Health
3.13. Access to Health Services
3.14. Other Health Impacts
Table 1 (line 147) presents relevant data although it is proposed to adapt the whole table in a single page to facilitate its reading.
Regarding the Discussion section (line 700), the important findings discussed show the need to study the psychosocial and mental health impact of such a vulnerable group as the persons experiencing homelessness. Hence, the authors present a relevant and current study, with broad practical implications that should be taken into account when addressing social policies.
Author Response
The article named “A Scoping Review of the Health Impact of the COVID-19 Pandemic on Persons Experiencing Homelessness in North America and Europe” is proposed for publication after minor (format) revision.
It is proposed a scoping review to delve into Covid-19 disease and homelessness as a central focus in a specific framework: North America and Europe. The authors justify the need to study this issue because the homeless population is one of the most vulnerable group to Covid-19, not only because of the potential infection and physical health problems, but also because of the psychosocial effect on their life context and mental health impairment it has had. As noted in lines 42-55:
“For example, the large-scale closure of public buildings and facilities reduced access to toilets and basic hygiene and sanitation for those who were unsheltered [11,12]. Less financial support from the general public due to stay-at-home advisories and closure or re-organisation of food and social support services also contributed to difficulties accessing food assistance and hygiene products (…)”.
In the Methodology section, the authors correctly reflect how the systematic search was conducted across five databases (EMBASE, Medline, CINAHL, PsychINFO, Web of Science) and Google Scholar (grey literature).
I really appreciate how they describe in Appendix A the search strategy. However, it is suggested that the authors present it in a figure/table format, to make it more visually appealing for the audience.
Thank you for this suggestion. We have adjusted the formatting and now present Appendix A as a table.
After the combined effort of 3 researchers, it was decided to select the articles according to the following criteria (lines 89-90; PEH= Persons experiencing homelessness):
“1) included primary data related to the impact of COVID-19 on health or well-being of PEH; and 2) were written in English”.
Besides, it is properly stated the exclusion criteria (lines 91-101). I would like to emphasize that articles prior to 2020 were excluded, so that they were effectively carring out an examination of the most recent and current context in this group.
Thank you for this comment. We have added expanded on our exclusion criteria and emphasized the reasoning for excluding articles published prior to March 2020, “The decision to exclude studies that defined history of homelessness as homelessness pre-dating 1 January 2019 was agreed upon by the research team to ensure that included studies were based on up-to-date housing data for the respective population. Additionally, as the World Health Organization declared COVID-19 a pandemic in March 2020 [32], publication in or after this month was deemed by the research team to be an appropriate requirement for included literature, in order to capture the health impacts of the pandemic on PEH.”
It is suggested though that, in future studies, the authors could include these previous articles in order to be able to make a comparison between their earlier and present life contexts after Covid-19. Also, another future line of research could be to compare the homeless population with other vulnerable populations or even the normalized population towards to explore differences and similarities in terms of the influence of Covid-19 on their health impact and psychosocial context. Perhaps even the differences between socioeconomically disparate contexts (countries) can be explored.
Thank you for these suggestions. We have included these as areas for future research within our limitations section.
Figure 1 shows PRISMA Flow Diagram in a very visual way that makes it easy to read and understand the procedure the researchers have followed.
The results are correctly collected and described. Besides, they are presented in an appropriate organization (starting in line 119):
3.1. Screening Process
3.2. Overview of Included Studies
3.3. SARS-CoV-2 Infection & Morbidity
3.4. COVID-19 Related Hospitalisation
3.5. COVID-19 Related Mortality
3.6. Fear of COVID-19
3.7. COVID-19 Vaccine Acceptance
3.8. Housing
3.9. Access to Personal Hygiene and PPE
3.10. Access to Food
3.11. Substance Use
3.12. Mental Health
3.13. Access to Health Services
3.14. Other Health Impacts
Table 1 (line 147) presents relevant data although it is proposed to adapt the whole table in a single page to facilitate its reading.
Thank you for this suggestion. We have adjusted the formatting so that Table 1 is presented on a single page.
Regarding the Discussion section (line 700), the important findings discussed show the need to study the psychosocial and mental health impact of such a vulnerable group as the persons experiencing homelessness. Hence, the authors present a relevant and current study, with broad practical implications that should be taken into account when addressing social policies.
Thank you for your thoughtful feedback and for considering our manuscript for publication.